# Planetary Health and Anthropocene Discourse: The Role of Muslim Religious Leaders

Mona Said El-Sherbini [1,2,3,4,*], Yusuf Amuda Tajudeen [4,5,6,7,*], Habeebullah Jayeola Oladipo [4,5,7,8], Iyiola Olatunji Oladunjoye [5,9], Aminat Olaitan Adebayo [7,10] and Jemilah Mahmood [11]

1   Narrative Medicine and Planetary Health, Integrated Program of Kasr Al-Ainy (IPKA), Faculty of Medicine, Cairo University, Cairo 11562, Egypt
2   Department of Medical Parasitology, Faculty of Medicine, Cairo University, Cairo 11562, Egypt
3   Invited Faculty, The Nova Institute for Health, Baltimore, MD 21231, USA
4   The Nova Network, Baltimore, MD 21231, USA; tajudeenamudayusuf@gmail.com
5   Department of Microbiology, Faculty of Life Sciences, University of Ilorin, P.M.B. 1515, Ilorin 240003, Nigeria; iyiolaoladunjoye@gmail.com
6   Department of Epidemiology and Medical Statistics, Faculty of Public Health, College of Medicine, University of Ibadan, Ibadan 200132, Nigeria
7   Planetary Health Alliance Campus Ambassador, Boston, MA 02115, USA; oladipohabeebullah@gmail.com
8   Faculty of Pharmaceutical Sciences, University of Ilorin, P.M.B. 1515, Ilorin 240003, Nigeria
9   Rouleaux Foundation, Suite 66, Agege 100283, Nigeria
10  Department of Agricultural Extension and Rural Development, Faculty of Agriculture, University of Ibadan, Ibadan P.O. Box 22133, Nigeria
11  Sunway Centre for Planetary Health, Sunway University, Petaling Jaya 47500, Malaysia; jemilah@sunway.edu.my
*   Correspondence: monas.elsherbini@kasralainy.edu.eg (M.S.E.-S.); ytajudeen2713@stu.ui.edu.ng (Y.A.T.); Tel.: +20-100-246-5704 (M.S.E.-S.); +234-(0)-706-206-3691 (Y.A.T.)

**Abstract:** The Anthropocene epoch marks a critical phase in the history of humanity, where anthropogenic activities have profoundly impacted our planet. Alongside remarkable ecological crises, the Anthropocene worldview has raised existential questions, with a cultural and ethical discourse that recognizes the intrinsic value and calls for more responsible sustainable living. Addressing these collective challenges necessitates a broader perspective guided by a unified sense of purpose toward personal and planetary health. In this context, the role of religious leaders in shaping the social and environmental worldviews of their followers cannot be underestimated. Religious teachings provide a moral framework for promoting climate action, global ethics, the rights of Indigenous peoples, peace, and justice, and other aspects of planetary health. By examining the global ecological crises through the lens of Islam, the Religion of Nature, or *Din al-Fitrah*, and its environmental and spiritual teachings, we can gain valuable insights into humanity's connection to the fabric of creation and its interaction with the world. These principles, rich in moral values, are intertwined with accountability and social cohesiveness. Therefore, the role of Muslim religious leaders considering the planetary-scale threats warrants further elucidation, recognizing that many other faiths and faith leaders can similarly contribute together for the common good.

**Keywords:** planetary health; muslim religious leaders; islam; Anthropocene; ecological crises

## 1. Introduction

The Anthropocene refers to a distinct geological epoch that addresses the scale and impact of human activity on Earth, to the extent that the planet is being altered due to human activities rather than natural causes [1]. This proposed epoch carries ethical implications, conveying a powerful message about collective human actions and their historical consequences [1]. From a scientific standpoint, the consequences of anthropogenic activities on human health and well-being are evident in the increased burden of non-communicable

diseases, emerging and re-emerging infectious diseases, poorer mental health, and even loneliness epidemics [2,3]. Additionally, exposure to extreme weather events such as heat-waves, wildfires, earthquakes, and hurricanes, as well as threats to food security, climate refugees, and unsustainability of human societies driven by rapid biodiversity loss and climate change crisis, pose a serious threat to global public health [2]. In 2015, the term '*planetary health*' was introduced into the academic lexicon by the final report of The Rocke-feller Foundation–Lancet Commission on planetary health. This initiative was developed to address the escalating threats posed by the Anthropocene era on deteriorating natural systems and their impact on human civilization, upon which it depends [2]. Planetary Health Alliance (PHA) defines planetary health as '*a solution-oriented, transdisciplinary field and social movement focused on analyzing and addressing the impacts of human disruptions to Earth's natural systems on human health and all life on Earth*' [4]. While the concept of plan-etary health may seem relatively new, its ideas and core principles are deeply rooted in ancestral origins [5]. Indigenous knowledge systems, informed by our ancestors, depict living knowledge embodied within the people who carry it forward and are intertwined with their ontological principles. Such a perspective also recognizes the planetary health message of the complex ecological interdependence and interrelatedness at micro-, meso-, and macro-scales [6,7]. A comprehensive and inclusive summary of planetary health, as reported by Prescott and Logan, emphasizes the intricate human relationship with the natural environment, its ancestral past, and its evolutionary resonance in the modern era, bridging the gap between intentions and behaviors while encompassing the notion of "hope" as a fundamental aspect [8].

In this paper, we contend that a successful transition from the Anthropocene should not be restricted to academic discourse, but also include a social movement that embraces spirituality and socio-cultural perspectives [9]. Kevin Schilbrack argues that "*The criterion for what belongs in the academy is not whether one's inquiries are value-laden—they always will be—but whether those values are open to challenge and critique*" [10]. From a systems perspective, drawing upon religious, indigenous, and traditional wisdom, the individual sphere plays a crucial role in shaping global leadership capacities for planetary health [10]. In 2023, Krause proposed four spheres in the handbook on Global leadership practices for planetary health, wherein leaders can influence the behavioral (action), collective (we space), and the larger macro-level environment (systems) [11]. *However, it is essential to recognize that the emphasis is often on the individual in a compartmentalized view, neglecting the spiritual dimension* [11]. Weir further argued that "*The impact on 'leadership' is that it is often difficult to separate the agency of individuals from that of wider collectives*" [12]. The significant role of religious leaders, as highlighted by the UN Secretary-General, António Guterres, bring people together to work for the common good. In 2015, his predecessor, Ban Ki-Moon, also stated that there is a profound responsibility to protect the fragile web of life on this Earth for the generations to come and it is time for the world's faith groups to collaborate with science [13]. Furthermore, the global call to spiritual faith leaders of all faiths by the Sao Paulo Declaration on planetary health emphasizes the moral dimension of protecting all life on health and leading in the creation of unity and solidarity [13]. Common themes found in world religions, such as stewardship, respect for life, ethical consumption, interconnectedness, compassion, and social justice, can serve as guiding principles in navigating the Anthropocene landscape [13,14]. In this context, Islam as a major world religion encompasses the human '*inner*' being and intentional actions aimed at promoting the well-being of the planet and all its inhabitants. Islam features a symbiotic unity found in diverse cultural paradigms and transformative schools of thought that inform leadership approaches across time and space [11,15]. Furthermore, the operational principle of societal obligations discourages all Muslims from remaining passive spectator in their community [16]. Instead, there is an inherent moral duty for Muslims to meet societal needs, whether urgent or more general [16]. This concept holds significant relevance in the context of the Anthropocene and planetary health discourse. By examining the global ecological crises through the lens of Islam, the Religion of Nature, or

*Din al-Fitrah*, and its environmental and spiritual teachings, we can gain valuable insights into humanity's connection to the fabric of creation and its interaction with the world. The focus of this viewpoint article is four-fold: first, to consider the theoretical delineation of the Anthropocene and related planetary crises; second, to probe the ecological and spiritual repercussions of the Anthropocene within a socio-cultural context; third, to engage planetary health discipline through the lens of Islamic sciences; fourth, to unfold the role of Muslim religious leaders in the era of the Anthropocene; finally, to provide avenues for future research studies and recommendations.

## 2. The Implications of the Anthropocene and Anthropocentrism

The Anthropocene epoch was initially a conceptual proposal coined by the Nobel Prize winner chemist, Paul J. Crutzen, in 2000 [17]. The Anthropocene implications have sparked much discussion and critique among various academics, poets, philosophers, politicians, and activists due to their highly ethical ramifications [18]. Rothe's article highlights several discourses that have emerged regarding the Anthropocene era, including *eco-catastrophism*, *eco-modernism*, and *planetary realism* [18]. Eco-catastrophism seeks to slow down human history on Earth by imposing '*planetary boundaries*' that illustrate the scenario of going above the 2 degree Celsius threshold. In this, eco-catastrophism calls for immediate international action needed to mitigate a dangerous shift in the Earth system—a new global form of '*Earth system stewardship*'. In contrast, Eco-modernism accepts human action on Earth and seeks to accelerate it, promoting the use of technology '*techno-environmentalism*', including climate geoengineering (intentional technological manipulation of Earth's climate), nuclear power, big data, artificial intelligence, and genetic engineering, which is claimed to be intelligently used to '*create a planet that is better for both its human and non-human inhabitants*' [18]. Planetary realism, on the other hand, acknowledges the Anthropocene and seeks to use it profoundly as an opportunity to foster a deeper understanding of the human earth's boundedness and work toward new approaches to understanding and cooperating with the planet Earth [18]. In this sense, it is thought that both natural and human systems are resilient, making it possible and legitimate to conduct technological experiments on the entire planet. However, this discourse also places a strong focus on Indigenous and traditional knowledge systems with its historical construct on interaction with nature. Nonetheless, the corresponding planetary experimentation still ought to be embraced as a geo-political project [18].

*Anthropocentrism*, the tendency to perceive the world through a human-centric lens, plays a significant role in the emergence of these multiple discourses [1]. This perspective is often reflected in the techno-centric models that dominate modern society [18]. From an indigenous wisdom perspective, in *Te Reo Māori*, the language of the Indigenous peoples of Aotearoa New Zealand, '*whenua*' means the land and people are one [19]. Redvers et al. stress on "*First Law values*", which played a role in promoting sustainable practices and resilient approaches to environmental and health-related challenges for millennia [20]. Indigenous people's concepts, such as viewing nature as *Mother Earth*, highlight the wisdom embedded in their portrayal of the Earth as a living organism sustaining life [20]. According to the Intergovernmental Science-Policy Platform on Biodiversity and Ecosystem Services (IPBES): *Mother nature is an expression used in a number of countries and regions to refer to the planet Earth and entity that sustains all living things found in nature with which humans have an indivisible, interdependent physical and spiritual relationship*. Other nature lexicons include *Terra*, *Mother Nature*, *The Blue Planet*, and *Gaia* [21]. Environmental activist Vandana Shiva eloquently captures this interconnectedness by stating the following: "*Our future is inseparable from the future of the Earth. It is no accident that the word human has its roots in 'humus', the Latin word for soil. Adam, the first human in the Abrahamic traditions, is derived from 'Adamah', meaning soil in Hebrew*" [22]. The African wisdom perspective of *Ubuntu* has similarities with Indigenous notions in North America, which can be translated as "*I am because we are*" [23], or "*I am a person by virtue of other persons*" [24]. As Elkington argued, "*This communal focus of Ubuntu has radical implications for how a person might relate to other persons, and to the planet*" [25].

So, what is required is an attitude of openness towards alternative ways of experiencing the self, and the world or what Boaventura de Sousa Santos calls the "*epistemologies of the South*" [26]. These general philosophical and sociological orientations rooted in indigenous beliefs contradict the Cartesian view of nature (nature as an inert object) [19–26]. Thus, by embracing diverse cultural and epistemological notions, new insights can be gained, the understanding of our very selfhood can be redefined, and actions oriented towards planetary health goals can be sparked, aiming to create a positive shared future for all.

## 3. The Intersection between Religion, Science, and Ecology

The word *science* originates from the Latin word *scientia*, which encompasses any system of knowledge. Science follows a systematic methodology, aiming to achieve a profound understanding of how the natural and social world functions [27]. However, since the seventeenth century, the prominence of modern and postmodern science has witnessed the marginalization of spiritual or metaphysical aspects. Likewise, the inclusion of an ontological component within medical and life sciences remains a subject of ongoing debate, with evidence-based measurable data often taking precedence [27]. Albert Einstein eloquently addressed this matter, expressing the view that "*Science without religion is lame, religion without science is blind*", a statement that emphasizes the potential for the mutualistic beneficial consideration of both scientific and religious perspectives [28]. An illustrative analogy can be drawn as follows: humans, with their limited knowledge, possess only a pixel, a tiny fragment of the bigger picture, while God, the supreme creative agency, has the entire picture, and humans are trying to connect its dots [29]. These ontological examples underscore the profound multi-faceted nature of science, which can greatly benefit from the incorporation of religious discourse, for a more holistic, inclusive, and rational perspective [27,28]. Thomas Berry also emphasizes the importance of understanding the relationship between the world's religions and the natural world, in the context of the entire planet and its integrity [30]. In the late 1960s, with the global environmental movements, Sayyed Hossein Nasr argued that the ecological catastrophe is intertwined with a spiritual crisis, wherein humanity's disconnection from the metaphysical has resulted in the destruction of the natural world [31]. Similarly, Fazlun Khalid points out that desacralizing the natural world has consequences for our spiritual state and our connection to God [32]. Ibrahim Ozdemir contends that humanity's alienation from nature is due to a loss of awareness of the divine quality of the environment and modern worldview that has rendered all philosophical claims and speculation useless [33]. This has led to a purely utilitarian view of nature, justifying its exploitation, the animals it nurtures, and its resources as well [1,33].

However, the ecological crises have promoted a shift towards viewing religion from a new perspective, an ecological perspective in which everything is interconnected as an organic whole [30,33]. In Pope Francis' encyclical *Laudato Si'*, an intriguing connection is made to the Sufi al-Khawwas, echoing the teachings of Rumi. They both affirm that the true essence of nature can be grasped by the mystical seeker [34,35]. This perspective, rooted in Qur'anic eco-philosophy, unveils the interconnectedness of all aspects of nature, forming a vibrant and living whole [35]. It can be likened to the "*great chain of being*" or *maratib al-wujud* [36], a concept shared in both Islamic and Western traditions, as poetically expressed by Alexander Pope [37]. This interconnectedness implies that the great chain of being is also a profound living miraculous "*great chain of consciousness*", a phrase aptly coined by Mohammed Rustom [38]. It suggests that consciousness permeates the entire spectrum of existence, from infinite to finite, leaving no aspect devoid of this inherent awareness [38]. This perspective blurs the artificial lines around the notion that nature is placed within set boundaries; rather, it focuses on synchronized harmony with a precise, delicate balance [1], a concept that resonates with Muslims' principles of *tawhid* (the interconnectedness of all creations toward one God), *taqwa* (living with God-consciousness), and *tasbih* (the constant state of God's glorification by all creatures that is invisible to the human senses and mind) [1,32].

Nevertheless, anthropocentrism, like other centric viewpoints, has created a stark division in how people perceive nature or the role that it plays in human affairs and the prevailing status quo [1]. While these ontological points of view are presented in a simplified manner, they can offer some clarification when discussing the Anthropocene with respect to the existential idea of the human relationship with the natural environment [1]. Faith-based traditions and religious scholars have the potential to infuse a spiritual and ethical dimension and participate in transformative social change [39]. Christian theologian Norman Wirzba further elaborates that viewing people from the standpoint of soil highlights the interdependence of humans and the planet [40]. The Holy Qur'an also states that humans are transcendent beings made of soil [41]. This perspective underscores the importance of recognizing that our health is closely tied to the health of the entire ecosystems on both micro and macro levels [7,41,42]. Since everything we consume relies on the soil and its nutrients, it would be a more accurate exemplification of ourselves as embodied and embedded beings [39] to view soil as a complex body containing billions of organisms and foundational microbes, including the holobiome and microbiome, enabling the daily cycles of biological processes within every living species in the vast ecological theatre on Earth [7,42].

These principles align strongly with the concept of planetary health, which implies that human health is planetary health [2,7,9]. Moreover, considering soil in relation to being human opens up the potential for rich interreligious dialogue, as the Earth serves as a metaphor for the unity of creation [14,20,22,34,39,40]. Overall, integrating scientific and religious perspectives can help elucidate the human relationship with the natural environment in the context of the Anthropocene.

## 4. Historic and Modern Insights on Islamic Sciences Significant to Planetary Health

The long intellectual history of Islamic sciences, as a system of knowledge, is derived from the Sacred book of the Qur'an and the hadith (record of Prophet Muhammed's actions and narrations, peace be upon him) [43]. The essence of Islam is centered upon the logical hierarchical order of nature that follows a vertical top-down approach with unity of all creations towards one God, the oneness belief [44]. Islamic sciences, place the heart as the spiritual intellect, with immense gratitude for the beauty, power, and intelligent mastery design of God [29]. All lifeforms and creatures from the cellular to the cosmic level are seen as a set of messages offering signs with a divine code that scientists are encouraged to decipher [45]. Islamic civilizations have made substantial advancements in various fields, including mathematics, science, medicine, philosophy, and astronomy [43]. Rich epistemological and dynamic cultural resources in Islamic studies resulted from the immense respect of other cultures, a supreme prophetic hallmark of traditional Islamic teachings [15]. During the Islamic Golden Age (8th–15th century), scholars built upon the knowledge of earlier civilizations, including the ancient Egyptian, Greek, Roman, Persian, Indian, Chinese, West African, and many others [43,46]. Through translation, the preservation and expansion of these grand narratives have taken place, coupled with the mutual synthesis of cross-cultural thinking [15,43,46]. Islamic civilizations fostered a rich tapestry of scientific, intellectual development, creative growth, novelty, and flourishing of the human spirit [43]. This cross-pollination of knowledge across different cultures and civilizations, including Greco-Roman, Judeo-Christian, Hinduism, and Confucianism [43,46] eventually influenced the making of the European Renaissance and the birth of modern science [43]. The brilliant works of literature, philosophy, and science with the philosophico-scientific conception of nature by great Muslim thinkers and influencers belong to Western traditions, like Ibn Rushd, Ibn Arabi, Avicenna (Abd Allah Ibn Sina, commonly known in English by his Latinized name: Avicenna), and Rumi (Jalal ad-Din Muhammad Balkhi, famously known as Rumi to the English-speaking world). The question remains as to why these *particular* works are elevated to the *universal* constellation of Western scientific intelligentsia while leaving behind other holistic and historic ecophilosophical works by Aziz al-Din Nasafi, Mahmud Shabistari, and Mulla Sadra [41]. Against this historic construct,

weaving together insights from diverse perspectives, historic and modern, can provide a comprehensive framework for human-nature interaction. In contemporary times and as the environmental and holistic health movements gain momentum, notable Muslim scholars like Sayyed Hossein Nasr have been widely recognized for their contributions in linking the pressing ecological crises to a spiritual crisis since the 1960s till date [31,44,47]. Also, Ziauddin Sardar and S.Parves Manzoor approached the environmental crisis from an Islamic science perspective during the 1970s and 1980s [48]. Over time, publications in the theoretical area of *Islamic eco-theology* and *environmental ethics* have gradually but steadily increased, and aspects of Muslim environmental activists have been encountered in the real world [49–55]. In this sense, the existence of a philosophical doctrine serves as a moral compass for people, including those seeking science [1]. As part of the same fabric of creation, Islamic teachings urge the conservation of nature and all its inhabitants [51]. Ozdemir further emphasizes that the Qur'an refers to the diversification of species, similar to human communities, recognizing the right of all species on earth to co-exist peacefully as established communities (*ummah*) and have their own languages, communication capacity, and respect for life [33]. It is considered that the perfect order of the whole nature, small and large, is constantly bringing us spiritual light to reflect on the beauty and power of its maker [44,45]. These Islamic principles promote a profound attitude of respect and preservation for biodiverse species and all lifeforms, biotic and abiotic alike, grounded in the moral obligation towards one God [55]. Thus, the ideas of ecological ethics and sustainability that are spiritually more profound are well documented in the writings of many Muslim scholars who addressed environmental challenges [31–33,48–55] and also in the *Islamic Declaration on Sustainable Development* [56], the Organization of Islamic Conferences (OIC) [57], and *a Muslim Declaration on Nature* [56] of the Muslim World League [48], to mention just a few.

While human's relationship to the rest of the creation has been addressed in Islamic science, the academic field known as Islam *and* planetary health is a fairly modern construct [58]. However, approaching Islam from a planetary health perspective can reveal a profound holistic ecology, recognizing the human–ecological–spiritual interactions as inter-relational in a unified system, i.e., a single creation [52,58]. These synergies aspire to infuse a spiritual, and hence ethical, engagement with earth and the biodiverse species inhabiting it, as well as the way we interact with the rest of the creation. As such, insights from Islamic teachings within a planetary health paradigm can illuminate a component of spiritual accountability, which is essential to counteract the status quo in contemporary societies.

## 5. Role of Muslim Religious Leaders

The Anthropocene represents a threat that is not only irreversible but also '*seemingly*' absolute in time and space [1]. It was even referred to by planetary health experts as the *Anthropocene syndrome* [59]. In a contemporary setting, Islam faith-based traditions are employed in the production of new meaning-making that conforms to prevailing power structures of the status quo, and notions of individualism and identity through consumption and materialism [11,16,53]. This also includes epistemological considerations and scientific studies that address climate and health, and the digital future [40,59,60]. Islamic studies with their profound orientation in the divine wellspring of knowledge can provide meaning to the past, present, and future, bridging the gap between science, technology, and spirituality [43,45,49,60–62]. Cultivating the true image of Islam as a religion that highly respects other cultures is crucial [15]. For example, Chinese Muslim culture commonly calls their faith *Qǐng Zhēñ Jiào* (the religion of the pure and real) instead of *yisilan Jiào* (the religion of Islam), a foreign-sounding to native Chinese ears. Also, the ideogram *Kāi tiān gu jiào* (the primordial religion from the world's beginning) is prominently inscribed upon entering a Chinese mosque [15]. This encounter reflects that Islam was not alien to their people's legacy but belonged to the very ethos of Ancient China, representing the best of its religious and philosophical traditions [15,46]. Such insights on how Islam clearly

reflects the distinctive bedrock (indigenous culture) over which they flow [15] are crucially important as globalization and knowledge validation seem to harmonize indigenous forms of cultural expressions in the Anthropocene era [11,15]. The Qur'anic teachings refer to the diversity of languages, skin color, and cultures and encourage humanity to "*get to know one another*" in a mutually beneficial way [63]. Thus, the role of Muslim religious leaders became crucial in our rapidly changing times, underscoring the critical role of human flourishing at the center for planetary health and world peace [16,58].

*5.1. Embracing Ecological Consciousness: A Spiritual Pro-Planetary Path*

In the context of the Anthropocene and planetary health discourse, the need to integrate a positive pro-planetary mindset is highlighted by many planetary health scholars [7–9,42]. A remarkable articulation of the role of a Muslim as a trustee, a 'Khalifa' with stewardship and guardianship, is charged with the responsibility of seeking, attaining, and cultivating knowledge in all domains of life to make the planet a better place for all beings [44,47]. The Qur'an mentions the word "Earth" 485 times and many verses on creation and humanity as part of a balanced whole [33,63]. For example, about 750 verses in the Holy Book of the Qur'an highlight the creation of the natural world, the laws that govern it, and its impact on human well-being and the sustainability of the environment [33,63]. Pope Francis has even acknowledged that non-Muslims can learn from the Holy Qur'an, as it is a book of many levels and great depths [45].

From a holistic perspective, the spiritual *consciousness* among all creatures can find the explanation in the linchpin Arabic term '*fitrah*' [60,64]. Fitrah represents the primordial covenant and the inherent goodness within human beings since they are born, which also signifies their intrinsic awareness as well as the natural disposition to incline towards the divine truth while also embracing morality [29,64]. Nurturing this innate affinity towards God's agency becomes a life-long process, centered on the purification of the heart [64]. Essentially, Islam, which is based on its root name "*salam*", which means peace (Religion of Peace), serves as a program to restore calmness to the heart and peace of mind through the remembrance of the divine qualities, thereby affirming Islam as the Religion of Nature, or *Din al-Fitrah* [60,64]. Moreover, the commitment to embracing mercy, harmony, and beauty within the spiritual heart stands as a fundamental and intrinsic quality in the context of Islamic tradition, one that is cultivated within the individual and extends outward to encompass all fellow beings and the entire spectrum of creation. This theological foundation forms the bedrock of Islam's identity as the Religion of Mercy. Expounding upon the extensive significance of embodying the mercy ethos has the potential to contribute substantially to the broader discourse on world peace, planetary health, and global well-being [65].

Islamic canon "*Shari'ah*" emphasizes the path to God, which encompasses more than just a set of higher principles and private spiritual practices. It is a dynamic system that provides meaningful guidance and a wealth of many contemporary environmental issues for the welfare of all created beings [37]. For example, within the scope of shari'ah, there is a specific chapter on environmental jurisprudence called *fiqh al-bi'ah*, with guidance on human actions related to the natural environment while stressing moderation in behavior [60]. By delineating and expounding on these concepts, incorporating God's message from the stream of previous prophets highlighted in the shari'ah, holistic environmental literacy, and ethics can be framed. Muslims hold the belief in all prophets, with Prophet Muhammed, the seal of the prophets, serving as a continuation, confirmation, and completion of the universal divine message [45,60].

Consequently, the role of Muslim religious leaders is distinctive in providing timeless wisdom from sacred sources, recognizing the extensive array of God's messages concerning the preservation of biodiverse species and all lifeforms [50]. Muslim religious leaders can also offer comprehensive reflection on action, including interpretations of how nature functions and insights into the sustainable and ethical use of natural resources within the spirit of Islam [65–67].

Furehaug emphasizes that ecological decline in the Anthropocene serves as a wake-up call, offering 'signs' to pause and rethink by bringing focus to ideas of free will, responsibility, and accountability for human actions and more opportunities for human agency [1].

> *'Corruption has appeared throughout the land and sea by (reason of) what the hands of people have earned so that He may let them taste part of (the consequence of) what they have done so that perhaps they will return (to righteousness)'.* (Qur'an 30:41) [63]

Even more intriguing is the corruption of the arth that will manifest itself in several ways, including soil erosion, ocean acidification, the destruction of food systems, wildlife extinction, and climate change [1]. These are also "*signs*" to contemplate and reflect to find the way back *(to righteousness)*, as explained by Furehaug [1]. In fact, one-eighth of the Qur'an exhorts Muslims to meditate on nature [63]. These Islamic perspectives are important reminders of their moral precepts about the natural systems of the Earth and the resources therein [1,31,32]. According to Hamza Yusuf Hanson, the president of Zaytuna liberal arts college in the USA, the cultivation of mindfulness and reflection with awe and wonder of the natural world has become critically important in today's modern era marked by technological distractibility with prevailing '*engineer's mindset*' that seems to fit everything in a linear mechanistic fashion [64]. Thus, traditional Islamic thought places the heart, rather than the brain, as the center of intellect, consciousness, and conscience. This understanding permeates all aspects of Islamic sciences, including the inward sciences that focus on the purification of the heart [64].

The modern campaign 'Reduce, Reuse, Recycle' aligns with religious environmental conscience enhanced by the concept of Islamic *Iktisad* (frugality). Prophet Muhammed practiced this principle in his usual way of mending his clothes and repairing his shoes, highlighting the value of mindful consumption and sustainability of resources within the practice of Islam [53]. Prophet Muhammed also encouraged the planting of trees (to 'hold onto *the sapling in your hand*'), even '*if the last day was established*'. The hadith emphasizes humanity's responsibility to preserve the natural environment no matter what condition prevails with a seemingly simple action of *'planting a sapling'*, yet the weight attached to the action is of great spiritual and ecological significance [60]. The Prophet Muhammed's sayings and positive actions give hope no matter what condition an individual is in, or the condition of the natural environment [60]. The prophet also considered the act of planting trees or sowing seeds as a charitable gift with a moral, ecological, and spiritual orientation, displayed inwardly and outwardly: "*If one plants a tree or sows seeds, if then a bird, a person, or an animal later eats from them, this action is regarded as a charitable gift (sadaqah) for him*" [51,58]. Therefore, every good action is considered of great spiritual significance, i.e., '*Actions are based upon intentions*', was stated by the Prophet Muhammed [60].

These examples of the common good shift the human consciousness in the mind to the re-centering of consciousness in the spiritual heart [60,64]. In an academic context, exploring the perspectives of Western non-Muslim intellectual elites regarding the Prophet Muhammed reveals intriguing insights. Alphonse de Lamartine, a renowned nineteenth-century French poet, articulated that the Prophet Muhammed created a spiritual nationality that united the people of every tongue and race through the restoration of rational dogmas [68]. Similarly, Sir George Bernard Shaw, the distinguished British playwright, referred to Prophet Muhammed as the "*Savior of Humanity*" [69]. In his influential book, *The 100: A Ranking of the Most Influential Persons in History*, Michael Hart positioned Prophet Muhammed as the foremost spiritual leader throughout human history while stating his global impact to serve humanity [70]. In contemporary times, the prominent scholar Dr. Craig Considine provides a sociological analysis of Prophet Muhammed's teachings and examples in his book *The Humanity of Muhammed, a Christian View*. Considine highlights Prophet Muhammed's wise situational leadership skills, forging peaceful co-existence in religious pluralism that aimed to uplift our common humanity, a side often forgotten in mainstream depictions and media narratives [71]. Thus, drawing inspiration from the Islamic worldview, the exemplary role model of Prophet Muhammed, and the moral teach-

ings of the Qur'an, can serve as a guiding lighthouse for personal and planetary health across time and space.

*5.2. Leveraging Planetary Health Message in the Virtual and Real Environment*

The urgency for collective action in the Anthropocene discourse is heightened by the assumptions of *a single future planetary-scale crisis* [2,4,7,9]. Notably, the shared emphasis on environmental protection for planetary health transcends diverse religious and global indigenous knowledge beliefs [5–7,11,19,20,22,30,34,41]. In the context of the Anthropocene discourse, Muslim leaders' social obligations, ethical responsibilities, and scientific roles become crucial. This is due to their potential to inspire proactive ecological solutions within the growing population of over 2 billion Muslims globally, who share a profound and meaningful common identity [11,16,47,48,58]. Muslims worldwide, despite their heterogeneous cultural and ethnic backgrounds, represent a cohesive dynamic community [15,16]. To this end, environmental initiatives should encompass phasing out greenhouse gas emissions, investment in net-zero carbon solutions, preservation of natural biodiversity, and prohibition of species exploitation [57,58,60,67]. Such actions will certainly reinforce the sacred message of 'planetary health' in the Holy Book of the Qur'an [58,63].

The holistic worldview embedded in Islamic teachings, as depicted through the storytelling style of the sacred Qur'an, has a profound potential to inspire learning and motivate concerned Muslims to act [58,60,63,66]. A pertinent example can be inferred from the story of Prophet Yusuf/Joseph, recounted in the Qur'an. During his time, he actively participated in devising a developmental strategy for the state of Egypt. Prophet Yusuf effectively addressed economic crises and climatic changes through a practical and innovative approach to ensure food security in times of adversity. This historic exemplary approach demonstrates the Qur'anic emphasis on proactive and sustainable solutions to address societal challenges [63].

Integrating Islamic environmental ethics, characterized by its non-anthropocentric nature, into modern educational curricula can foster and enable goal-oriented actions synergic with planetary health [58,60,66,67]. Collaborative efforts with organizations like the United Nations Educational, Scientific, and Cultural Organization (UNESCO) in projects such as the "Qur'anic Botanic Garden" promote sustainable gardening and plant conservation education [71,72]. Moreover, the United Nations initiative of World Interfaith Harmony Week represents a valuable opportunity for advancing planetary health [72]. By facilitating interfaith dialogue and collaboration, this initiative provides avenues to discuss shared values and foster collective actions toward a sustainable and resilient future for our planet [14,35,42]. In contemporary pluralistic societies, there exists a pressing need to revitalize historic intercultural knowledge exchange for planetary health, reminiscent of the Islamic golden age's contribution to the European Renaissance, bridging the gap between East and West. In the digital era, this exchange transcends traditional social and physical boundaries [11,15,43,48,73–75]. In the USA, the Garrison Institute provides an example of bringing together spiritual thought leaders with academic sciences and humanities to promote actions for planetary health [76]. This effort, and many others, align with the collective goal of enhancing collaboration and effective dialogue between diverse faith-based communities and scientific communities to reinforce the value of spiritual perspectives in addressing the profound challenges of the Anthropocene.

Muslim religious leaders bear a significant responsibility in facilitating development policies and practices aligned with any development goals. This task necessitates considering various factors, such as the interconnectedness of religious and cultural norms, local relevance, national sovereignty, security concerns, and the needs of marginalized indigenous communities [77]. For instance, the Muslim worldview challenges the Neo-Malthusian argument of population control [78] without adequately addressing the root causes of materialistic and deterministic logic that perpetuate inequalities between advantaged and less advantaged communities [7]. Such principles are acknowledged in

the context of planetary health [7], which also emphasizes the moral understanding of sustenance (*risq*) within the Muslim perspective [16].

Dr. Umar F. Abdullah, a prominent American Muslim scholar, presents a holistic and moderate interpretation of Islamic teachings, thoughts, and practices that offer valuable insights into the context of planetary health. He draws inspiring examples of women's legacies in Islam from models of communal living and regenerative agriculture [15,16,78,79]. His local and global initiatives serve as an exemplary manifestation of Islam that embraces an open-minded approach while anchoring spiritual principles, integrating knowledge of diverse cultural heritages, and critically examining ecological solutions for the common good [80].

To disseminate knowledge and promote an environmental vision and practices while leveraging planetary health messages, key Muslim stakeholders are encouraged to organize in-person and virtual eco-Islamic conferences and engage with global Imams, local mosque committees as well as traditional Islamic scholars [56,57,60]. Global and local Islamic leaders and scholars should also collaborate with different faith-based communities to foster intrinsic values and meaningful inter-religious dialogues aimed at addressing ecological and spiritual crises [34,41,67]. For example, the Islamic perspective clearly shows synergies with other religions and cultures, such as the Buddhist concept of *upaya*, which accommodates the teaching of the 'truth'. In a sense, *scientia sacra* contains both the '*seed and the fruit*' for what is viewed as the '*tree of knowledge*' [80,81]. Also, the Japanese traditional culture places a strong emphasis on the concept of "*sensei*", literally translated as 'teacher' but goes beyond that to encompass mastery of knowledge and character development. This theme can be similarly encountered in the example of Khadr and Prophet Moses representing the ideal relationship between the teacher and the student represented in classical Sufi books and the Qur'an [82].

Islamic teachings provide a significant opportunity to encourage environmental action with accountability and infuse spiritual consciousness toward planet health, as Islam is inseparable from the natural world, and environmental care is an integral part of its faith [37,40].

On the other hand, a multidisciplinary approach to planetary health encourages local and global synergies between scholars of philosophy, metaphysics, natural and social science, neuroscience, and indigenous knowledge and religious leaders who can help to reinvigorate a metaphysical spiritual view of creation and the natural environment [9,19,30,31,38,58,72]. Relying solely on science and technology is arguably insufficient to address the current complex ecological crises in the Anthropocene while sidelining the metaphysical component [1,27,37,38]. For example, the gap between science and spirituality is exemplified by the concerns instilled in Western communities regarding the absence of a human–spiritual dimension in artificial intelligence, with a humanoid robot like Mindar seen as a new deity of the future, in reality, trained on human data [61,83]. The concept of '*blessed algorithms*' and the critical flaws that arise when trained on biased human data highlight the need for caution and awareness of the magnitude of an anthropocentric worldview dominated by an engineer's mindset and materialistic profiteering [84–86]. Muslim religious leaders and scholars must continue to provide a meaningful moral compass in the face of ecological and spiritual crises in the synthetic age [31,85,86].

Taking inspiration from the original Alliance of Virtue (*hilf al-fudul*) established by Prophet Muhammed (peace be upon him) in the 7th century Common Era, the New Alliance of Virtue seeks to promote a shared commitment to peace, human dignity, and mutual understanding between all people irrespective of race, ethnicity, or religion, thus, serving as a model for contemporary action in the digital age and for the common good [86]. Islamic leaders and scholars have historically played a vital role in advocating for sustainable Islamic core principles, and their importance cannot be overstated [43,47,49,56,57,65,87]. Thus, Islam's binocular vision can provide a strong foundation for promoting active sustainable values while acknowledging the role of human agency in contemporary ecological and spiritual crises.

Furthermore, the Qur'an is believed by Muslims to be an all-time eco-educational program of life and living marked by moderation, concern for equity, and free of waste [58,63].

## 6. Avenues for Future Research Studies

To explore avenues for potential future research studies, it is imperative to consider the rich contextual themes of planetary health from an emic, inside-out perspective compatible with the Islamic worldview and teachings. This perspective aligns with the integral theory model, which underscores effective global leadership practices for planetary health. This model involves the enhancement of self-leadership by an inquiry-based process that facilitates the mapping of transformative changes from individuals to collective to macro-level spaces, as presented by Krause in their research design [11].

To harmonize with the holistic approach of planetary health and the discourse around *connectedness to nature* [42]; in 2004, Mayer and McPherson Frantz formulated a measure for the personality trait connectedness to nature. This trait reflects the emotional bond individuals feel as they connect to the natural world [88,89]. However, a study by Perrin and Benassi study in 2009 suggests that the connectedness to nature scale primarily captures cognitive belief rather than emotional connection to the natural environment [90]. Psychometric properties and correlations of cognitive belief with related variables such as empathy, social connectedness, and connectedness to nature, encompass more than transient situations, feelings, or moods; they represent a fundamental attitude toward life [90]. In response to the need for spirituality measures applicable across religious traditions, a psychometric scale was developed to capture *oneness belief* [90–92]. The positive correlations observed between personal life satisfaction and oneness belief provide a platform for exploring numerous new research questions concerning the role of Muslim religious leaders in advancing personal and planetary health [90]. Similarly, Islam, like other religions, places significant emphasis on environmental concerns, encompassing preservation and development [90]. Garfield et al. 2014 have established an empirical connection between oneness beliefs and pro-environmental attitudes and behaviors [92]. Recent studies (Affandi et al. 2022; Abdul Hamid and Abd Matalib 2021; Bsoul et al.2022) have delved into various prospects of Islamic ideology and principles concerning environmental behavior [72,93–95]. In this context, a Value–Belief–Norm model (*Al-Mizan*) has been proposed, rooted in the Qur'an teachings, to establish an environmental-based ethos that amplifies balance and harmony. This theoretical framework encourages the achievement of the critical balance between the spiritual, societal, and material needs of humanity. The al-Mizan concept promotes pro-environmental engagement and proactive behavior to live in harmony with self, with others, and with nature and everything in it [96–98].

The advancing notion of *human flourishing* has implications for both personal well-being and planetary health, as underscored by Logan et al. in 2023 [99]. However, correlations with *individual cognitive beliefs* as well as *social connectedness* and *connectedness to nature* can offer a comprehensive analysis for deeper comprehension of how these frames influence individual and collective environmental behavior and actions [90]. For example, McFarland et al., 2012, developed a scale for measuring a person's closeness and supportive behavior towards humanity, endorsement of human rights, and responses in favor of global harmony, without regard for race, religion, tribalism, group boundaries or other distinctions [100]. "*The identification with all humanity*" or the IWAH scale offers an operational measure of the degree of individual expression with known-groups validation [100]. Additionally, future research should probe into the concept of teleology from the perspectives of Islamic teachings, which assert that the world does have a purpose, as indicated in the Qur'an [63,100].

On the scale of environmental sustainability research, the case study of *Permaculture* serves as an example of holistic positive attitudes and behaviors toward sustainability. This approach integrates land, people, resources, and the environment using symbiotic beneficial synergies [41]. Permaculture with its indigenous roots in sustainable eco-friendly agricultural and land management practices aims to enrich Earth's biodiversity, a goal

congruent with the Qur'anic ecophilosophy of human stewardship towards all life forms. Iskandar Waworuntu and Dr. Umar F. Abdullah exemplify this Islamic Permaculture ethos. Though they reside in different parts of the world, namely Indonesia and America, their mutual engagement in Islamic Permaculture in the Bumi Langit farm in Indonesia unites them [40]. This example reveals the tenets of the great chain of consciousness with awareness of the intricate interconnectedness of all beings, both biologically and spiritually, a notion amplified and holds relevance in Islamic teachings [38,41].

From an academic perspective, the integration of rigorous interdisciplinary studies of Indigenous Knowledge Systems (IKS), ecological and spiritual perspectives within planetary health domains is paramount [11,19,20,25,26,31,80]. This calls for incorporating ancient knowledge and wisdom within the global knowledge benchmarks, orchestrating different worldviews, and leveraging resources like the Oxford Islamic Studies databases. For example, a rigorous Cochrane Review is imperative to systematically evaluate empirical evidence spanning global Islamic history, leadership, and cultural paradigms [101]. Evaluating a culture's efficacy involves coherence, consistency in behavior, and spirituality nurtured within a shared prosperity framework [15,101]. Such endeavor amalgamates the intricate tapestry of IKS, fostering religious perspectives, historical evaluation, and cultural assessment, which can enrich our understanding of the past and guide our path forward [15].

## 7. Recommendation

As part of the effort towards pursuing a sustainable future, we urge active engagement with faith-based leaders, recognizing and appreciating their influential role in guiding the communities toward the planetary health path. By encouraging interdisciplinary collaborations among public health experts, scientists, religious scholars, and environmentalists, we can improve our expertise and effectively address the complex planetary health crisis of the present-day world. Moreover, fostering the development of educational programs and initiatives centered on faith-based perspectives to address the complex environmental and health challenges can raise awareness within the religious communities (such as the Muslim communities) about the intricate connection between faith and environment, whereas raising awareness on sustainable practices within these populations will facilitate positive change and enhance pro-environmental behaviors. Consequently, the exploration of interfaith collaborations is a perfect way to avoid silo thinking while encouraging unity among diverse voices in a shared commitment to ensuring the sustainability of the planet. For a lasting impact to be achieved, incorporating the perspectives, ideas, and opinions of the religious leaders in the policy-making process and implementation is essential, ensuring that religion and science remain mutual and work in tandem. Future research should focus on the interpretation of and application of the sacred texts from the Qur'an, conducting comparative studies across other religions for the sole aim of identifying the similarities and distinctions in religious teachings concerning the need to protect the environment and actions taken by faith-based leaders. Conducting an assessment of the long-term impact of religious leaders' involvement in awareness creation about the sustainability of the environment especially within the Muslim communities' concomitant with the analysis of their role in shaping environmental policies is an important step to gain a comprehensive understanding of the dynamics of influence. Finally, the establishment of a global planetary health network or improving existing planetary health networks with the active involvement of Muslim religious leaders is highly crucial in facilitating collective efforts to protect the planet. Such global networks have the propensity of serving as a bridge between faith-based wisdom and policy that would be taken on the environment on a global scale. Beyond this, such global networks will further contribute to a more inclusive approach to planetary health, thereby reflecting the diversity of perspectives while promoting cross-cultural dialogues that transcend beliefs.

## 8. Conclusions

This article emphasizes the importance of engaging religious leaders, including Muslim leaders, in promoting environmental protection, ethical values, and planetary health. The authors call for collaboration between scholars from various disciplines and spiritual leaders to address ecological and planetary health challenges. The article also highlights several areas for potential research related to planetary health and its relationship with Islamic sciences and perspectives, covering various aspects such as integral theory models, cognitive beliefs, well-being, teleology, permaculture, and indigenous knowledge systems. These research areas do contribute to a deeper understanding of how cognitive frameworks influence environmental behaviors and actions. Overall, these research areas can contribute to the development of knowledge regarding a sustainable approach to planetary health and its relationship with Islamic sciences and other perspectives. Such research can help understand how cognitive beliefs, culture, and ancient wisdom influence human actions related to environmental protection and planetary health. These thought processes, which are more expansive than the prevalent but constrained discourse on pressing issues like climate change and environmental degradation, seek to inspire a spiritual awakening, inter-religious dialogue, and, as a result, ethical engagement with the earth, the species that inhabit it, and the rest of creation in the web of life.

**Author Contributions:** Conceptualization, M.S.E.-S., Y.A.T. and H.J.O.; methodology, M.S.E.-S., Y.A.T. and H.J.O.; resources, M.S.E.-S., Y.A.T. and H.J.O.; data curation, I.O.O. and A.O.A.; writing —original draft preparation, M.S.E.-S., Y.A.T. and H.J.O.; writing—review and editing, J.M.; supervision, M.S.E.-S. and J.M. All authors have read and agreed to the published version of the manuscript.

**Funding:** This research received no external funding.

**Data Availability Statement:** Information available in this manuscript is solely intended for educational and research purposes and does not endorse any specific beliefs, practices, and personalities mentioned within. Readers should consult relevant authorities, scholars, professionals, and experts for additional resources when conducting related research. The authors and publishers of this manuscript are not responsible for any outcomes that may occur from the application of the information in this manuscript.

**Acknowledgments:** We extend our deep appreciation to Rianne C. Ten Veen for her insightful expertise and suggestions in enhancing the quality of this manuscript. Her comprehensive knowledge of sustainable development and intercultural dialogue has greatly added to the value of our research, specifically, in the context of planetary health and interconnectedness. We also extend our appreciation to the editor of Challenges Journal for their time and consideration towards publishing this manuscript. To the anonymous reviewers, we appreciate the suggestions and comments provided to improve the quality of this manuscript.

**Conflicts of Interest:** The authors declare no conflict of interest.

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
