# Peer review of "Planetary Health and Anthropocene Discourse: The Role of Muslim Religious Leaders"

_challenges, doi:10.3390/challe14040046_

Round 1
Reviewer 1 Report (New Reviewer)
Comments and Suggestions for Authors
This topic is very important and timely. I think a few modifications will improve clarity, especially for non-Muslim readers.
-The structure of the paper is a bit confusing. I recommend you have a literature review section and a recommendations section. Right now these items are mixed in different sections and it is challenging to follow.
-Clearly faith traditions have much to offer the planetary health movement, but how have they contributed to the harm done to people and planet in the past. Why do modern religions offer something different?
-The paper talks about the role of faith leaders in the abstract, but very little content is about strategies that faith leaders can use.
-The last section before the conclusion seems to depart from the previous content. It uses old citations and doesn't effectively flow. Consider an earlier placement if used.
-There are some terms that need to be defined so all readers understand what is being said. For example Planetary Health Alliance and Religion of Nature or Din al Fitrah.
-Overall this work reviews the Muslim faith's stance on the environment, but I think it could push even further through use of the San Paulo Declaration and the global call to faith leaders.
Thank you for this work. These modifications will ensure the message reaches a broader audience.

Author Response
Comments and Suggestions for Authors,
This topic is very important and timely. I think a few modifications will improve clarity, especially for non-Muslim readers.
Dear Reviewer,
The authors would like to thank you for the comments and valuable suggestions provided to further improve the quality of the paper and also enhance the clarity and relevance of our work for a broader audience. This is how we have addressed the comments point-by-points.
Comment: The structure of the paper is a bit confusing. I recommend you have a literature review section and a recommendations section. Right now these items are mixed in different sections and it is challenging to follow.
Response: Regarding the structure, we have reorganized the paper to provide a more coherent and comprehensive overview of existing scholarship related to the topic. Additionally, we have created a separate recommendations section to highlight actionable suggestions more explicitly in line 625- 657 of the revised manuscript.
Comment: Clearly faith traditions have much to offer the planetary health movement, but how have they contributed to the harm done to people and planet in the past. Why do modern religions offer something different?
Response: We have improved the section that delves deeper into this aspect, elucidating on different approaches to address these issues in lines 324-326, 359-366 , 540-542 and 584-586 of the revised manuscript.
Comment: The paper talks about the role of faith leaders in the abstract, but very little content is about strategies that faith leaders can use.
Response: We have expanded the section discussing strategies and actions that faith leaders can undertake. This addition aims to provide practical insights into their potential roles in promoting planetary health.
Comment: The last section before the conclusion seems to depart from the previous content. It uses old citations and doesn't effectively flow. Consider an earlier placement if used.
Response: We have ensured a smoother transition in the penultimate section, aligning it more cohesively with the preceding content to maintain the overall flow and logical sequence of ideas.
Comment: There are some terms that need to be defined so all readers understand what is being said. For example, Planetary Health Alliance and Religion of Nature or Din al Fitrah.
Response: We have included concise definitions for terms such as Planetary Health Alliance, Religion of Nature, and Din al Fitrah to ensure a better understanding for readers from diverse backgrounds in lines 350 to 359.
Comment: Overall this work reviews the Muslim faith's stance on the environment, but I think it could push even further through use of the San Paulo Declaration and the global call to faith leaders. Thank you for this work. These modifications will ensure the message reaches a broader audience.
Response: Your suggestion to further explore the San Paulo Declaration and the global call to faith leaders is insightful. We have integrated references to these influential documents to enrich our exploration of the Muslim faith's perspective on environmental stewardship and its connection to global initiatives in the lines 96-98 of the revised manuscript.
We genuinely appreciate your critical insights, which are instrumental in refining and expanding the depth and relevance of our work for a wider audience.
Thank you.
Sincerely,
Authors.

Reviewer 2 Report (New Reviewer)
Comments and Suggestions for Authors
Dear Authors,
The paper entitled "Planetary Discourse on Health and the Anthropocene: The Role of Muslim Religious Leaders" appears to be intriguing due to its philosophical and religious aspects. However, from my perspective, the paper must meet the basic requirements of the journal.
Detailed Comments:
1.
The Abstract should be structured, including the background of the study, clear research, objectives, methodology, results, and a general conclusion.
2.
The introduction to the research is properly presented; however, at the end of the introduction, the authors should state a clear research objective instead of research tasks, or propose an alternative research hypothesis to the null hypothesis for subsequent verification.
3.
Methodology is not defined.
4.
Results
A. The results presented in the manuscript discuss general philosophical and sociological orientations rooted in the beliefs of indigenous peoples, which contradict the Cartesian concept of nature (nature as an inanimate object).
A. Embracing diverse cultural and epistemological conceptions can bring new perspectives, redefine our understanding, and inspire actions towards planetary health goals, aiming to create a positive shared future for all.
B. The perspective on the relationship between humans and the rest of creation is discussed in Islamic sciences, but the field of Islamic sciences and planetary health is a relatively modern construct. However, approaching Islam from the perspective of planetary health can reveal a deep ecological holism that recognizes human-ecological-spiritual interactions as mutually interdependent within one integrated system - one creation. These synergistic aspirations aim to enrich spiritual engagement and, therefore, ethical engagement with the Earth and the diverse species that inhabit it, as well as how we interact with the rest of creation. This way of looking at Islamic teachings within the planetary health paradigm can shed light on the spiritual dimension that is essential to counter the current status quo in contemporary societies.
C. The Islamic canon "Sharifah" emphasizes the path to God, which includes more than just a set of higher principles and private spiritual practices. It is a dynamic system that provides significant guidance and many environmental issues aimed at the well-being of all created beings.
D. In contemporary pluralistic societies, there is an urgent need to revive historical cross-cultural exchange in the context of planetary health, reminiscent of the Islamic Golden Age's contribution to the European Renaissance, among others. In the digital age, this exchange transcends traditional social and physical boundaries. The Garrison Institute in the United States acts as a catalyst, supporting spiritual thought leaders in utilizing the potential of faith to promote actions for planetary health. The actions of this and other social institutions support the common goal of strengthening cooperation and dialogue among diverse faith communities, enhancing the significance of spiritual perspectives in addressing the deep challenges of the Anthropocene.
E. A multidisciplinary approach to planetary health encourages local and global synergies between scholars of philosophy, metaphysics, natural and social sciences, neuroscience, and indigenous knowledge and religious leaders who can help to reinvigorate a metaphysical spiritual view of creation and the natural environment. Muslim religious leaders and scholars must continue to provide a meaningful moral compass in the face of ecological and spiritual crises in the synthetic age.
F. Taking inspiration from the original Alliance of Virtue (established by Prophet Muhammad in the 7th century Common Era, the new Alliance of Virtue seeks to promote a shared commitment to peace, human dignity, and the common good, serving as a model for contemporary action in the digital age.
I. Islam and environmental care: The author emphasizes that Islam is inseparable from the natural world, and environmental care is an integral part of this faith. Environmental actions in line with the scientific and ethical messages of Islam are important tools in pursuit of planetary health.
J. Multidisciplinary approach to planetary health: The author suggests that a multidisciplinary approach is crucial in pursuit of planetary health. Scholars from philosophy, metaphysics, natural sciences, social sciences, neuroscience, indigenous knowledge, and religious leadership can collectively contribute to revitalizing a metaphysical and spiritual view of creation and the natural environment.
K. Balanced approach to science and spirituality: The author highlights the existing gap between science and spirituality in Western societies and concerns about the absence of a human-spiritual dimension in artificial intelligence. This emphasizes the need for caution and awareness of the magnitude of an anthropocentric worldview dominated by an engineer's mindset and materialistic profiteering.
L. Pursuit of the common good: The authors stress that Muslim religious leaders and scholars play a crucial role in promoting the balanced core principles of Islam and should continue to provide a significant moral compass in the face of ecological and spiritual crises.
M. Example of the Alliance of Virtue: The author provides an example of the original Alliance of Virtue established by Prophet Muhammad in the 7th century. The new Alliance of Virtue aims to promote a shared commitment to peace, human dignity, and the common good, serving as a model for contemporary action in the digital age.
CONCUSION
In conclusion, the text emphasizes the importance of engaging religious leaders, including Muslim leaders, in promoting environmental protection, ethical values, and planetary health. The authors call for collaboration between scholars from various disciplines and spiritual leaders to address ecological and planetary health challenges.
The text also highlights several areas for potential future research related to planetary health and its relationship with Islamic sciences and perspectives, covering various aspects such as integral theory models, cognitive beliefs, well-being, teleology, permaculture, and indigenous knowledge systems. These research areas can contribute to a deeper understanding of how cognitive frameworks influence environmental behaviour’s and actions.
Overall, the text suggests that there are many research areas that can contribute to the development of knowledge regarding a sustainable approach to planetary health and its relationship with Islamic sciences and other perspectives. Such research can help understand how cognitive beliefs, culture, and ancient wisdom influence human actions related to environmental protection and planetary health.
Comments on the Quality of English Language
Minor English editing is required
Author Response
Dear Reviewer,
The authors would like to thank you for the comments and valuable suggestions provided to further improve the quality of the paper.
Your detailed observations and suggestions on structuring the Abstract, defining a clear research objective, and outlining the methodology are incredibly helpful. Additionally, your breakdown of the results and their implications on various philosophical and societal dimensions offers a valuable perspective.
We deeply appreciate your attention to the intricacies of the paper, especially in highlighting the importance of engaging religious leaders, particularly Muslim leaders, in environmental advocacy and promoting ethical values for planetary health. Your encouragement for interdisciplinary collaboration between scholars from diverse fields and spiritual leaders resonates with the core message of our work.
The guidance provided on potential areas for future research, particularly concerning the relationship between planetary health and Islamic sciences, as well as the influence of cognitive frameworks on environmental behaviors, is immensely insightful. These suggestions will undoubtedly enrich the depth and breadth of the discourse.
We are genuinely thankful for the time and effort you dedicated to reviewing our perspective article. Your constructive feedback will significantly contribute to enhancing the quality and depth of our work. We are committed to addressing each of the outlined points meticulously to ensure the paper aligns with the journal's requirements and maintains academic rigor.
Once again, thank you for your invaluable input and support.
Sincerely,
Authors.

This manuscript is a resubmission of an earlier submission. The following is a list of the peer review reports and author responses from that submission.
Round 1
Reviewer 1 Report
Comments and Suggestions for Authors
This appears to be a narrative literature review but there is no description of methodology or search terms which might have shaped the review. Therefore, it is not clear that the conclusions have a scientific basis even if they have many ethical, sociological, political and religious justifications. The paper might read better as a commentary or editorial indicating that these are personal views rather than new research.
While it is true that engagement with Muslim Religious Leaders on Sustainability and the Anthropocene may bring benefits, it is more important to engage with communities on the science and ethics of human impact on the planet. This cannot be left to religious leaders alone. Indeed, some of the challenges in sustainability, not least the need for educational and employment opportunities for girls and women and the benefits of birth control, have not always been helped by formal religious doctrines.
The paper refers to both Islamic Science and to the Science of Islam but without defining either concept. I am not sure this terminology is helpful as scientific methods and thinking are distinct from religious beliefs. There is a whole field of literature on the benefits of tolerant historic Islamic regimes on the protection of Greek and Roman texts and their development of mathematical and scientific discovery during the European dark ages but none of that literature implies that Islamic science differs from classical science or renaissance science (nor from Judeo-Christian or Hindu or Confucian science).
Comments on the Quality of English Language
I noticed an expression ‘Anthropocene marking’ in line 96. This is not an expression I am familiar with and I wondered if this is a typographical error?
Author Response
Dear Reviewer,
Thank you for taking the time to review our manuscript. We greatly appreciate your insightful suggestions and comments, which have significantly strengthened our manuscript.
Kindly find below the point-by-point revision addressing your comments:
Comments and Suggestions for Authors
This appears to be a narrative literature review but there is no description of methodology or search terms which might have shaped the review. Therefore, it is not clear that the conclusions have a scientific basis even if they have many ethical, sociological, political and religious justifications. The paper might read better as a commentary or editorial indicating that these are personal views rather than new research.
Response: This paper is indeed a viewpoint article., and we have made it clear in the revised manuscript.
While it is true that engagement with Muslim Religious Leaders on Sustainability and the Anthropocene may bring benefits, it is more important to engage with communities on the science and ethics of human impact on the planet. This cannot be left to religious leaders alone.
Response: This has been addressed in line 268-290 and line 309-317.
Indeed, some of the challenges in sustainability, not least the need for educational and employment opportunities for girls and women and the benefits of birth control, have not always been helped by formal religious doctrines.
Response: We have addressed this point in the revised manuscript in line 352-378.
The paper refers to both Islamic Science and to the Science of Islam but without defining either concept. I am not sure this terminology is helpful as scientific methods and thinking are distinct from religious beliefs.
Response: We appreciate your concern regarding the terminology used. In the revised manuscript, we have provided clear definition and explanation in line 129-142.
There is a whole field of literature on the benefits of tolerant historic Islamic regimes on the protection of Greek and Roman texts and their development of mathematical and scientific discovery during the European dark ages but none of that literature implies that Islamic science differs from classical science or renaissance science (nor from Judeo-Christian or Hindu or Confucian science).
Response: We have addressed this concern in the revised manuscript in line 187-204.
Comments on the Quality of English Language
I noticed an expression ‘Anthropocene marking’ in line 96. This is not an expression I am familiar with and I wondered if this is a typographical error?
Response: We apologize for the confusion caused by the expression “Anthropocene marking”. The conceptual proposal of ‘Anthropocene marking’ has been clarified in line 96 and its reference has been included in the reference list.
Thank you once again for your valuable suggestions. We have carefully addressed each point raised, and we believe that the revised manuscript was significantly improved.
Sincerely,
The Authors

Reviewer 2 Report
Comments and Suggestions for Authors
The article could use some additions. As it is, there is very little "new" information in the article. The article is not wrong in any of its thoughts or conclusions, but they are not new conclusions. Specifically connecting discourse on the Anthropocene to Islamic thought on the environment is laudable, but it would be good if the article were able to add something to the scholarship on Islam and the environment more than just this.
Comments on the Quality of English Language
Generally the language is fine, but I did notice that one Qur'an was spelled with the apostrophe and other times was not. With the apostrophe is more correct, but at a minimum this should be standardized.
Also, there is Qur'anic verse sighted, but there is no need to mention that this is a translation.
Author Response
Dear Reviewer,
Thank you for dedicating your time to review our manuscript. We greatly appreciate your insightful suggestions and comments, which have significantly strengthened our work.
Kindly find below the point-by-point revision addressing your comments:
Comments and Suggestions for Authors:
The article could use some additions. As it is, there is very little "new" information in the article. The article is not wrong in any of its thoughts or conclusions, but they are not new conclusions. Specifically connecting discourse on the Anthropocene to Islamic thought on the environment is laudable, but it would be good if the article were able to add something to the scholarship on Islam and the environment more than just this.
Response: In the revised manuscript, we have made substantial additions to our viewpoint. We have provided a comprehensive overview of the historical significance of the Islamic Golden Age, highlighting its contributions to various scientific disciplines. Additionally, we have incorporated a forward-looking exploration into the future, offering insights into the evolving role of Muslim religious leaders and highlighting the relevance of planetary health through the lens of Islam. Moreover, we have introduced the concept of a new Alliance of Virtue, which adds a unique and thought-provoking dimension to the discourse on planetary health in our manuscript. These additions contribute new perspectives to the existing scholarship on Islam and the environment.
Comments on the Quality of English Language
Generally the language is fine, but I did notice that one Qur'an was spelled with the apostrophe and other times was not. With the apostrophe is more correct, but at a minimum this should be standardized.
Response: We apologize for the inconsistency in the spelling of “Qur’an”. In the revised manuscript, we have ensured that the spelling is standardized throughout, using the appropriate form with the apostrophe consistently.
Also, there is Qur'anic verse sighted, but there is no need to mention that this is a translation.
Response: In the revised manuscript, we have omitted the mention of translation for the Qur’anic verse, as it is commonly understood that any citation of the Qur’an in an English-language article would be a translation. Thank you for pointing that out.
Thank you for your thoughtful review our manuscript. Your insightful suggestions and comments have, indeed, strengthen our manuscript.
Sincerely,
The Authors

Round 2
Reviewer 1 Report
Comments and Suggestions for Authors
I am afraid that this does not read like a scientific paper. It is not just that there is no methodology described and no results reported. More fundamentally, there appears to be a confusion between scientific enquiry and religious belief.
I can understand the case for examining the impact of religious leaders (positive or negative) on the outcomes of the Anthropocene era. One could also make a case for selecting Islam as a case study to develop tools to explore the potential impacts.
However, the concept of 'Islamic science, as a system of knowledge, is derived from the Holy book of Qur’an and the hadith' does not invite peer review and I think the paper consistently confuses science with belief. If one is a believer in God (as I am), one does not rely on scientific enquiry to confirm that belief: while advances in science might influence the context in which we believe and our understanding of how God might shape the planet or the universe, we do not rely on science to confirm our beliefs. If we did, we would not be believers.
This could be published as an opinion piece but it may not be suitable in this form for a peer-reviewed scientific journal.
Comments on the Quality of English Language
I think there is some loose use of English in this paper. One example is the use of the term lens, as in 'lens of Islam', 'lens of Islamic sciences', 'planetary health lens', 'anthropocentrism lens'. While a planetary health lens might be interpreted as a view based on published evidence of threats to planetary health and the interactions between risks to planetary boundaries, I am not clear how the other lens quotes should be viewed.
The abstract included the phrase 'value-leden principles'. I think this probably means value-laden principles but it is unclear if this is referring to religious principles or principles of good science (the two are not the same).